# Transcriptome Analysis of Halotolerant *Staphylococcus saprophyticus* Isolated from Korean Fermented Shrimp

**DOI:** 10.3390/foods11040524

**Published:** 2022-02-11

**Authors:** Eunhye Jo, Sungmin Hwang, Jaeho Cha

**Affiliations:** 1Department of Integrated Biological Science, Pusan National University, Busan 46241, Korea; joeunhye13@pusan.ac.kr; 2Clean Energy Research Center, Korea Institute of Science and Technology, Seoul 02792, Korea; sungminhwang@kist.re.kr; 3Department of Microbiology, Pusan National University, Busan 46241, Korea; 4Microbiological Resource Research Institute, Pusan National University, Busan 46241, Korea

**Keywords:** *Staphylococcus saprophyticus*, transcriptome, salt tolerance, Korean fermented product, jeotgal, saeu-jeotgal

## Abstract

Saeu-jeotgal, a Korean fermented shrimp food, is commonly used as an ingredient for making kimchi and other side dishes. The high salinity of the jeotgal contributes to its flavor and inhibits the growth of food spoilage microorganisms. Interestingly, *Staphylococcus saprophyticus* was discovered to be capable of growth even after treatment with 20% NaCl. To elucidate the tolerance mechanism, a genome-wide gene expression of *S. saprophyticus* against 0%, 10%, and 20% NaCl was investigated by RNA sequencing. A total of 831, 1314, and 1028 differentially expressed genes (DEGs) were identified in the 0% vs. 10%, 0% vs. 20%, and 10% vs. 20% NaCl comparisons, respectively. The Clusters of Orthologous Groups analysis revealed that the DEGs were involved in amino acid transport and metabolism, transcription, and inorganic ion transport and metabolism. The functional enrichment analysis showed that the expression of the genes encoding mechanosensitive ion channels, sodium/proton antiporters, and betaine/carnitine/choline transporter family proteins was downregulated, whereas the expression of the genes encoding universal stress proteins and enzymes for glutamate, glycine, and alanine synthesis was upregulated. Therefore, these findings suggest that the *S. saprophyticus* isolated from the saeu-jeotgal utilizes different molecular strategies for halotolerance, with glutamate as the key molecule.

## 1. Introduction

Several traditional methods are commonly used to preserve food in Korea. For example, jeotgal, a type of Korean fermented seafood product, is produced by salting whole fish, internal organs of fish and shellfish, including shrimp (saeu-jeotgal), anchovy (myeolchi-jeotgal), or pollock roe (myeongranjeot) [1]. The jeotgal is consumed as a side dish or used as an ingredient for making kimchi [2]. The unique taste and flavor of the jeotgal is produced during fermentation under high salinity (20–30% *v*/*v*) by the microorganisms present in seafood; for the saeu-jeotgal, the salt concentration is generally 35–40% [1]. Representative fermentation microorganisms include several *Staphylococcus*, *Halomonas*, and *Lactobacillus* species, which were previously isolated from the jeotgal using culture-based methods [3]. Recent advances in the next-generation sequencing technology revealed that *Tetragenococcus*, *Lactobacillus*, and *Staphylococcus* are also the predominant species in some jeotgal products [4]. These microorganisms facilitate the fermentation process, producing several peptides, aromatic compounds, and functional nutrients in the preserved seafood. Several studies have already investigated the mechanism of microbial resistance to salt [5,6]; however, the exact mechanism is still unknown.

In general, microorganisms exposed to high salt concentrations respond by moving water through the semipermeable membrane via osmotic pressure, in which the cells rapidly discard ions or organic solutes by opening the mechanosensitive channels (MSCs). To maintain homeostasis, microorganisms can utilize two strategies known as “salt-in” and “compatible solutes” [7]. The salt-in strategy is conducted through the accumulation of potassium ions (K^+^) in the cell against the high concentration of surrounding sodium ions (Na^+^). If homeostasis is not achieved, the microorganisms can synthesize or actively uptake the small molecules called compatible solutes, which are osmoprotectants that include polyols, sugars, amino acids, and quaternary amines (e.g., glycerol, trehalose, proline, glutamate, ectoine, and glycine betaine) [8]. These two strategies have been well-characterized in several microorganisms using genome, transcriptome, and metabolome data analyses.

Transcriptome analysis is conducted to identify the genes involved in stress response. Although a microorganism possesses genes for stress response, a specific gene may not be actively expressed as RNA unless cells are exposed to severe growth conditions. RNA sequencing (RNA-seq) is a useful technology to investigate the stress response mechanisms in various microorganisms [9,10,11,12]. For example, transcriptome analysis of *Staphylococcus aureus* revealed the increased cell adhesion and biofilm formation in response to low concentrations of ampicillin [13]. In addition, the mechanism of adaptation by *Staphylococcus xylosus* against osmotic stress in salted meat was explored using transcriptomics, demonstrating that the expression levels of several genes involved in glucose and lactate catabolism and glutamate synthesis was upregulated [14]. Furthermore, the transcriptome and metabolome analyses of *Tetragenococcus* revealed that glycine betaine was the major compatible solute under the high-salinity condition [6]. Transcriptome analysis of *Staphylococcus* sp. OJ82 isolated from fermented squid (ojingeo-jeotgal) showed that the expression of genes associated with compatible solutes, transporters, and cell membranes was upregulated under salt stress [15].

This study aimed to investigate the molecular mechanism underlying salt tolerance in the *Staphylococcus saprophyticus* strain isolated from saeu-jeotgal using RNA-seq and transcriptome analysis. The findings of this study provide additional knowledge on the biological strategies employed by halotolerant bacteria in salted and fermented foods.

## 2. Materials and Methods

### 2.1. Strain and Culture Conditions

*S. saprophyticus* was obtained from Prof. Do-Won Jeong of Dongduk Women’s University, Seoul, Republic of Korea. *S. saprophyticus* was isolated from saeu-jeotgal and grown in a tryptic soy broth (TSB) medium at 30 °C. The overnight culture was inoculated (1% *v*/*v*) in 300 mL TSB medium without salt (0% NaCl), and the cells were grown at 30 °C with shaking (200 rpm). When the optical density (OD) at 600 nm reached 1.3 after 6 h, all cultures were harvested using centrifugation at 6000× *g* for 30 min and then suspended in 6 mL TSB medium. For the experiment, equal volumes of cells were inoculated in 100 mL TSB media containing 0%, 10%, and 20% NaCl (*n* = 3 per condition) to simulate salt stress. Cell growth was monitored every 3 or 6 h, and the cells were harvested after a 6 h salt shock using centrifugation at 15,900× *g* for 3 min. To accurately measure the cell density, diluted cells (1/5) were measured and the calculations for calibration were followed.

### 2.2. RNA Extraction, Library Construction, and Sequencing

Cell lysis was conducted using acetone–ethanol (1:1 *v*/*v*), followed by resuspension using a TE buffer (10 mM Tris and 1 mM ethylenediaminetetraacetic acid (EDTA)). Further lysis was performed through the addition of 5 μL lysostaphin (1 mg/mL) using Lysing Matrix B (MP Biomedicals, Santa Ana, CA, USA) and homogenization for 20 s using a Mini-Beadbeater-24 (BioSpec Products Inc., Bartlesville, OK, USA). Total RNA was extracted using an RNeasy Mini Kit (Qiagen, Valencia, CA, USA) according to the manufacturer’s instructions. DNase I (Thermo Scientific, Waltham, MA, USA) treatment was performed to remove the DNA. The purified RNA was eluted with double distilled water. The RNA samples were sent to Macrogen (Seoul, Republic of Korea) for RNA-seq. The quality and integrity of the samples were checked using a 2100 Bioanalyzer (Agilent Technologies, Santa Clara, CA, USA). RNA samples with RNA integrity number > 9.0 were used for library construction using a NEBNext Ultra™ II RNA Library Prep Kit for Illumina (New England BioLabs, Ipswich, MA, USA). Sequencing was conducted for 100 bp paired-end reads using a HiSeq 2500 platform (Illumina, San Diego, CA, USA), generating 19–26 million reads per sample. The transcriptomic data were deposited in the Sequence Read Archive (SRA) of the National Center for Biotechnology Information Search (NCBI) database under accession code PRJNA797916.

### 2.3. Transcriptomic Data Analysis

The quality of the raw sequencing reads was assessed using FastQC v.0.11.9 [16]. Low-quality reads and adapter sequences were removed using Trim Galore v.0.6.6 [17] and Cutadapt v.3.4 [18]. The reference genome NZ_CP031196.1 and gene annotation files were directly downloaded from the NCBI database (https://www.ncbi.nlm.nih.gov/genome/?term=NZ_CP031196.1, accessed on 17 September 2021). The preprocessed reads were aligned to the reference genome sequence using Bowtie2 v.2.4.1 [19]. SAMtools v.1.13 was used to convert the SAM files into BAM files for downstream analysis [20]. The number of reads that mapped to each transcript was determined using HTSeq v.0.13.5 [21].

### 2.4. Differential Expression and Functional Enrichment Analyses

Differential expression analysis for the pairwise comparison of the treatment conditions was performed using DESeq2 v.1.32.0 [22]. The differentially expressed genes (DEGs) were identified by screening the genes with adjusted *p*-values < 0.05 and absolute log_2_ fold change (FC) values > 1. To quantify the DEGs, EnhancedVolcano v.1.10.0 [23] and ggvenn v.0.1.9 [24] were used. The eggNOG-mapper v.2.1.6 tool [25] was used for the Cluster of Orthologous Groups (COG) analysis, whereas ggplot2 [26] and dplyr [27] packages were used in R version 4.1.1 (R Foundation for Statistical Computing, Vienna, Austria) for data visualization.

## 3. Results

### 3.1. Growth of S. saprophyticus in Different NaCl Concentrations

*S. saprophyticus* was cultured in a TSB medium containing 0%, 10%, and 20% NaCl to mimic the salinity in jeotgal products. Generally, the sharp increase in salinity leads to a strong decline in cell growth rate. However, in the present study, the bacterial cells survived even in 20% NaCl condition, suggesting that *S. saprophyticus* is a halotolerant microorganism. After a 6 h treatment, the mean OD at 0%, 10%, and 20% NaCl was 8.01 ± 0.073, 5.44 ± 0.065, and 2.16 ± 0.035, respectively (Figure 1). Through the growth assay, we hypothesized that a salt tolerance mechanism in *S. saprophyticus* enables resistance to salt stress.

### 3.2. Overview of the Transcriptome Analysis

To elucidate the mechanisms underlying salt tolerance in *S. saprophyticus*, RNA-seq was performed using the total mRNA isolated from cells in the exponential growth phase after NaCl shock. RNA-seq data analysis was carried out with the more than 19 million reads from each sample, and approximately 92–97% of the reads were aligned to the *S. saprophyticus* reference genome (NZ_CP031196.1). DEGs analysis revealed that the treatments with 0%, 10%, and 20% NaCl induced significantly different gene expression patterns with low variations in the *S. saprophyticus* samples. The transcriptomic data from three biological replicates per treatment condition were obtained, and the expression patterns were visualized as a heatmap and a principal component analysis plot (Figure 2A,B). The variances of the first and second principal components were 71% and 27%, respectively, implying that the main factor to distinguish the transcriptomic dataset is the different concentration of salt. A total of 831, 1314, and 1028 DEGs were identified in the pairwise comparisons for 0% vs. 10% NaCl, 0% vs. 20% NaCl, and 10% vs. 20% NaCl, respectively. These DEGs represented approximately 32.5%, 51.4%, and 40.2% of the total number of predicted genes (2555) in the 0% vs. 10% NaCl, 0% vs. 20% NaCl, and 10% vs. 20% NaCl comparisons, respectively. A total of 305 DEGs were found to be common among the three treatment conditions, while there were 60, 199, and 108 specific DEGs identified in the 0% vs. 10% NaCl, 0% vs. 20% NaCl, and 10% vs. 20% NaCl comparisons, respectively (Figure 2C).

In addition, volcano plots were generated to examine the patterns of upregulated or downregulated DEGs among the treatment conditions (Figure 3A–C). The results revealed that the number of upregulated DEGs in the 0% vs. 10% NaCl comparison was higher than that of downregulated DEGs, and there were more downregulated DEGs than upregulated in the 0% vs. 20% and 10% vs. 20% NaCl comparisons (Figure 3D).

### 3.3. Functional Enrichment Analysis

The COG classification of the DEGs was determined using the annotation data for pathogenic *S. saprophyticus* because the genome sequence and protein annotation data of nonpathogenic *S. saprophyticus* are not available. Although the *S. saprophyticus* strain isolated from the jeotgal is different from the pathogenic strain, the DNA identity at the alignment and mapping steps was found to be reliable (92–97%).

The identified DEGs were classified based on the existing COG categories (Figure 4). The top five categories included “amino acid transport and metabolism” (373 DEGs), “transcription” (258 DEGs), “inorganic ion transport and metabolism” (252 DEGs), “carbohydrate transport and metabolism” (234 DEGs), and “translation, ribosomal structure, and biogenesis” (217 DEGs). Among these, “amino acid transport and metabolism” and “inorganic ion transport and metabolism” contained several DEGs encoding amino acids and ion transporters involved in the synthesis of compatible solutes and ion efflux proteins. As the salt concentration increased, the expression of genes involved in amino acid metabolism was upregulated, whereas the expression of genes related to ion transport was downregulated.

### 3.4. Analysis of Membrane Transporter Proteins

To predict the survival strategies of *S. saprophyticus* against salt stress, halotolerance-related genes were classified and compared with membrane transporter and compatible solute synthesis genes. The expression levels of the genes encoding stress response proteins, including ion transporters, were also compared (Table 1). As expected, the expression of genes encoding universal stress proteins (USPs) was upregulated in all the conditions. Notably, one gene encoding a USP (DV527_RS05825) was upregulated by 5.89-fold in the 0% vs. 20% NaCl comparison. By contrast, the MSCs responsible for water efflux and the maintenance of cell integrity, including a gene encoding the large conductance MSC protein MscL (DV527_RS07575), were downregulated under high-salinity conditions. Furthermore, expression of almost all the genes encoding symporters and antiporters, except sodium/glutamate symporter (DV527_RS02905) and dicarboxylate/amino acid symporter (DV527_RS05020), was downregulated. The transporter mediating the uptake of dicarboxylate/amino acid is known for the uptake of succinate, fumarate, and l-malate in bacteria [28]. The expression of genes encoding Na^+^ antiporters and K^+^ transporters, which are involved in the primary strategy for salt homeostasis, was downregulated at high salt concentrations. The expression of genes encoding betaine/carnitine/choline transporter (BCCT) family proteins that deliver small solutes into the cell was also highly downregulated. However, several genes encoding major facilitator superfamily (MFS) proteins that are responsible for the movement of various substrates were upregulated. Overall, these results imply that *S. saprophyticus* delivers various unknown substrates into the cell membrane to survive hyperosmotic environments.

### 3.5. Pathway Analysis of Compatible Solutes 

Table 2 summarizes the transcriptomic data of the genes involved in the metabolic pathways of compatible solutes. Although ectoine is a known compatible solute in bacteria, ectoine synthesis- or transport-related genes were not detected in the *S. saprophyticus* strain isolated from the jeotgal. Compared to the expression levels of the genes coding for ion transporter proteins, the genes related to amino acid synthesis, specifically osmoprotectants, were significantly upregulated. The expression of the gene encoding alanine dehydrogenase (DV527_RS05830) increased 1.01- and 4.05-fold in the 0% vs. 10% NaCl and 0% vs. 20% NaCl comparisons, respectively. The final products synthesized by alanine dehydrogenase were predicted to have been converted by the alanine–glyoxylate aminotransferase family protein (DV527_RS05740), thereby forming glycine and pyruvate [29]. For glycine synthesis, the expression of genes encoding glycine cleavage system protein P (GcvPB), GcvPA, GcvT, glycine C-acetyltransferase, and choloylglycine hydrolase family protein was also upregulated. By contrast, the expression of the gene encoding glycine glycyltransferase FemX (DV527_RS03365) was downregulated, likely because the enzyme is involved in peptidoglycan synthesis using glycine. However, the expression levels of the genes encoding Mur synthetases (MurC and MurD) and RacE were upregulated, consequently promoting peptidoglycan synthesis, which is usually elevated under the salt stress condition [30]. MurC, MurD, and RacE commonly utilize glutamate as the substrate, which is a key molecule for halotolerant mechanisms in *S. saprophyticus*. The gene encoding l-pyrroline-γ-carboxylate dehydrogenase (PruA), which is involved in the synthesis of proline from glutamate, was upregulated 6.02-fold in the 0% vs. 20% NaCl comparison. Hence, the accumulated glutamate under salt stress may be used both as a compatible solute and as a substrate to produce other compatible solutes. In this study, the upregulation of one FMN-binding glutamate synthase family protein (DV527_RS02350) and three poly-γ-glutamate hydrolase family proteins (DV527_RS12045, DV527_RS07840, and DV527_RS11825) induced intracellular accumulation of glutamate. This was confirmed by the reduced expression of genes encoding proteins that synthesize arginine (Arg) and poly-γ-glutamate (PGA) using glutamate as the substrate, such as Arg biosynthesis protein J (ArgJ), ArgB, PGA synthesis protein PgsB, and PgsC.

## 4. Discussion

The results from the growth assay suggest that *S. saprophyticus* is a halotolerant organism, not halophilic (Figure 1). In general, the growth rate of halotolerant bacteria decreases as the salt concentration increases. By contrast, the growth rate of halophilic bacteria, which prefer a saline environment, decreases when the salinity exceeds the optimal range. Several studies have been conducted to determine the adaptation mechanism of halotolerant and halophilic bacteria in salted foods. For example, the *Tetragenococcus* species in highly salted and fermented foods have been analyzed at the pan-genome, transcriptome, and metabolome levels [6]. *S.*
*saprophyticus* strains have been isolated from various salted and fermented foods, including sausages, soy sauce, soy paste, and cheeses; however, studies on the mechanism underlying halotolerance in this species are limited [31,32,33,34].

The salt stress response generally consists of a primary strategy for the adjustment of K^+^ and Na^+^ fluxes and a secondary strategy for maintaining a sustained response using compatible solutes. If the ion content is insufficient to limit water efflux, the microorganism sustains the stress response through the synthesis, import, and accumulation of compatible solutes [35,36]. The salt stress response mechanism of *S. saprophyticus* from the jeotgal is presented in Figure 5, which only demonstrates the response of *S. saprophyticus* after 6 h of NaCl treatment and not the overall osmoadaptation mechanism. The sampling time is critical because the growth stage affects the transcriptome analysis. In this study, the highly downregulated DEGs in *S. saprophyticus* were mainly associated with the primary strategy, such as MSC, K^+^, and Na^+^ transporters. Na^+^/H^+^ antiporters and K^+^ voltage-gated channels are known as the components of the primary strategy for halotolerance in *Staphylococcus* species [37,38]. In general, Gram-positive bacteria generally maintain high concentrations of K^+^ within their cells [7]. In the case of Na^+^ transporters, the expression of almost all the genes related to proteins including symporters and multi-subunit Na^+^/H^+^ (Mnh) antiporters was downregulated. Notably, the Mnh protein plays an important role in maintaining halotolerance [37]. In addition, the expression of the genes encoding BCCT channels was downregulated, implying that *S.*
*saprophyticus* may prefer different types of osmolytes. Among the membrane transporters, the levels of osmoprotectant ABC transporter substrate-binding proteins and USPs were upregulated. The ABC transporters are important components of the osmoprotectant import system. Furthermore, the OpuB and OpuC transporter systems have been well-characterized in *Bacillus subtilis*, with substrate specificities that are determined by substrate-binding proteins (e.g., OpuBC and OpuCC) [39]. Among the amino acid transporters, the dicarboxylate/amino acid:cation symporter (DAACS) and the sodium/glutamate symporter were upregulated. The DAACS family proteins are transporters for the uptake of dicarboxylates in bacteria, including succinate, fumarate, and malate [28,40]. In *Escherichia coli*, the DAACS protein was discovered to have an affinity for glutamate, subsequently confirming that the glutamate/aspartate-proton symporter has substrate-binding and transport functions [41]. Thus, the DAACS family protein observed in *S. saprophyticus* may also be involved in the transport of compatible solutes. Similarly, the expression of the gene encoding the sodium/glutamate symporter was elevated, indicating that glutamate was actively imported by *S. saprophyticus* under salt stress.

Although the accumulation of compatible solutes requires high amounts of energy, this strategy is generally effective in the adaptation to osmotic stress [42]. As well as the transporters, changes in the expression of the genes involved in the synthesis and degradation of amino acids were remarkable (see Figure 5). The accumulation of glycine betaine and carnitine was previously characterized in the *Staphylococcus* species, and several genes involved in glutamate synthesis were found to be upregulated [14,43,44]. Similarly to other Gram-positive bacteria, the expression of glutamate synthesis-related proteins was increased in *S. saprophyticus* [45,46]. The synthesized glutamate is used either as a precursor for proline by l-glutamate γ-semialdehyde dehydrogenase or as a material for peptidoglycan synthesis. The expression of the genes associated with glycine and alanine synthesis from the intermediate molecules in the tricarboxylic acid cycle was also upregulated. This implies that amino acids with small molecular weights are produced and, subsequently, used for inducing the mechanism for salt tolerance in *S. saprophyticus*. The roles of glutamate and other amino acids, which have been highlighted in salt stress, are well-matched to the unique taste and flavor characteristics of salted fermented fish.

Overall, the findings suggest that *S. saprophyticus* maintains osmotic homeostasis through the accumulation of compatible solutes to adjust to the high salt concentration after 6-h of salt stress. However, these results only represent the biological state of *S. saprophyticus* across the sampling time set for this study. Thus, additional research confirming the mechanism for halotolerance at the timepoint immediately after salt stress is required.

## 5. Conclusions

The *S. saprophyticus* isolated from saeu-jeotgal was investigated using RNA sequencing to determine the mechanisms for osmoadaptation. The isolated strain, which was cultivated in media containing different concentrations of NaCl, maintained its growth even at 20% NaCl. The transcriptome analysis revealed that the DEGs comprised 30–50% of the total number of genes across the pairwise comparisons for 0% vs. 10% NaCl, 0% vs. 20% NaCl, and 10% vs. 20% NaCl. The functional enrichment analysis revealed that the expression of DEGs classified under the “amino acid transport and metabolism”, “transcription”, “inorganic ion transport and metabolism”, “carbohydrate transport and metabolism”, and “translation, ribosomal structure, and biogenesis” categories was highly differentially expressed. Specifically, the genes encoding MSCs and Na^+^ transporters were downregulated, while those encoding for USPs were upregulated. Furthermore, several proteins related to amino acid synthesis and transport were upregulated. Notably, the expression of the genes related to glutamate, an important molecule for halotolerance in bacteria, was also elevated. Therefore, this study reports the role of glutamate as a key compatible solute and presents the potential molecular mechanism underlying the adaptive response of *S. saprophyticus* to salt stress. Based on this knowledge, this research provides valuable information for the development of starter strains for the fermented food industry.

## Figures and Tables

**Figure 1 foods-11-00524-f001:**
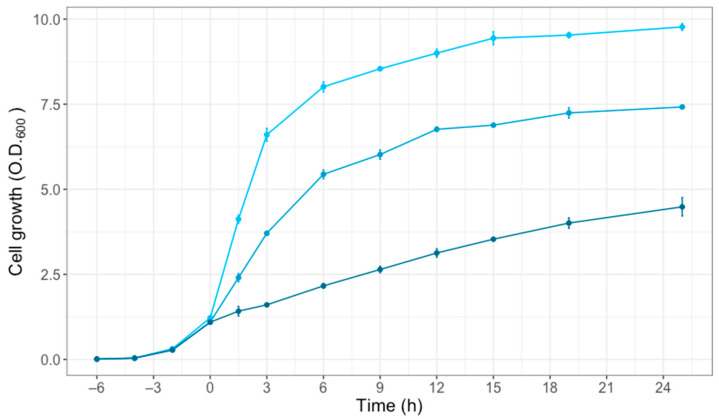
The growth of *S. saprophyticus* under different salt concentrations: 0% (light blue), 10% (blue), and 20% (dark blue) NaCl. The growth assay was conducted to determine the mean optical density at 600 nm (O.D._6oo_) for 24 h using three biological replicates. The error bars represent the standard deviation of the means.

**Figure 2 foods-11-00524-f002:**
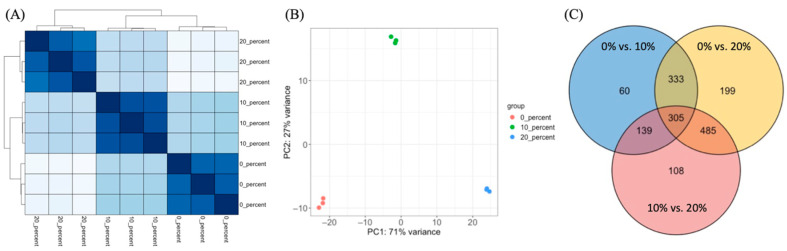
Overview of the transcriptome analysis results. (**A**) Heatmap and (**B**) principal component analysis (PCA) plot of the overall gene expression. (**C**) Venn diagram representing the number of common and unique differentially expressed genes (DEGs) per pairwise comparison: 0% vs. 10% NaCl, 0% vs. 20% NaCl, and 10% vs. 20% NaCl.

**Figure 3 foods-11-00524-f003:**
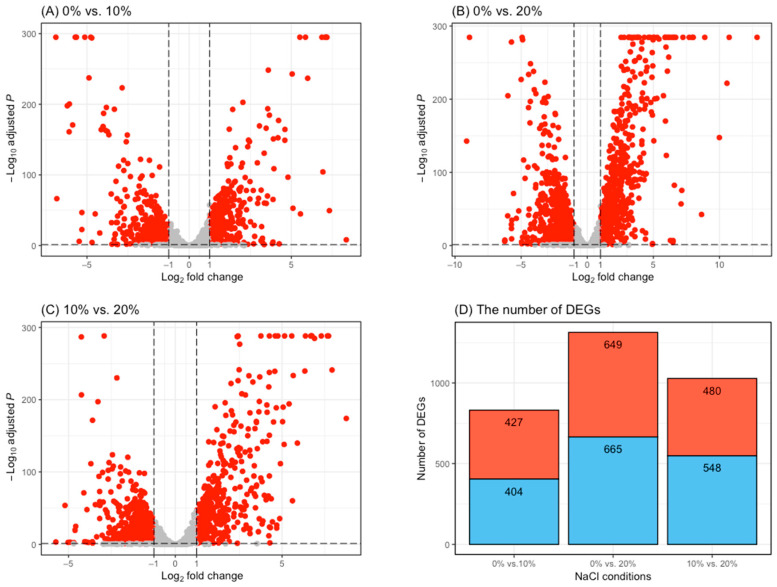
The volcano plots show the proportion of differentially expressed genes (DEGs) with adjusted *p*-values < 0.05 and absolute log_2_ fold change values > 1 (red dots) per pairwise comparison: (**A**) 0% vs. 10% NaCl; (**B**) 0% vs. 20% NaCl; and (**C**) 10% vs. 20% NaCl. (**D**) Bar graph showing the number of upregulated (red) and downregulated (blue) DEGs per pairwise comparison: 0% vs. 10% NaCl, 0% vs. 20% NaCl, and 10% vs. 20% NaCl.

**Figure 4 foods-11-00524-f004:**
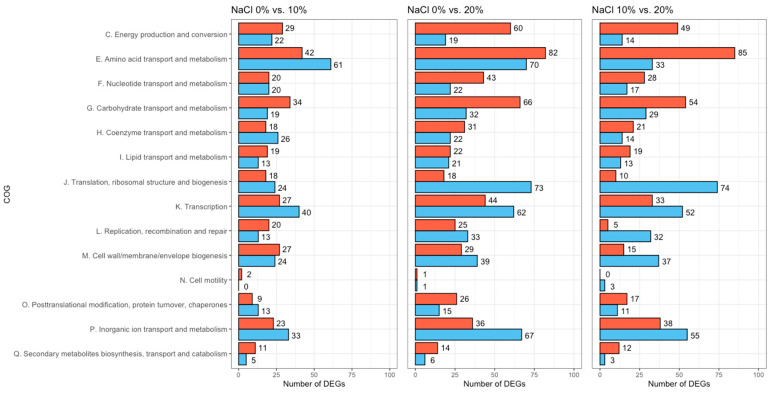
Cluster of Orthologous Groups (COG) classification of the identified differentially expressed genes (DEGs) per pairwise comparison: 0% vs. 10% NaCl, 0% vs. 20% NaCl, and 10% vs. 20% NaCl. The red and blue bars represent the upregulated and downregulated DEGs, respectively.

**Figure 5 foods-11-00524-f005:**
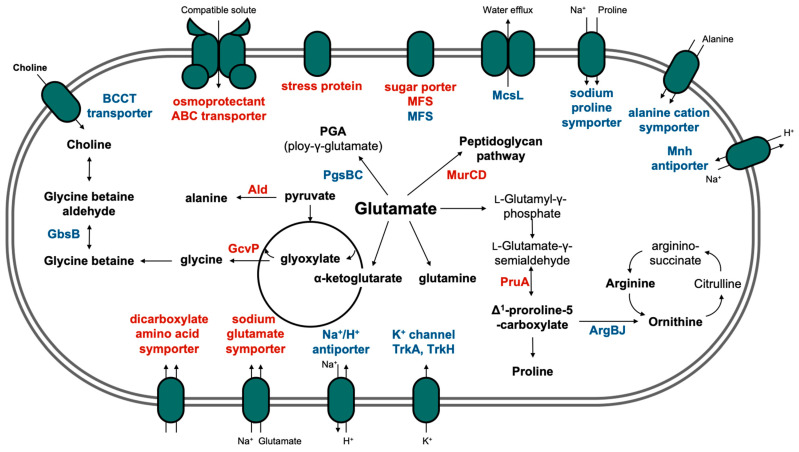
Schematic diagram of the metabolic pathways associated with the osmoadaptive response of *S.*
*saprophyticus* to salt stress (6 h NaCl treatment). The genes in the red and blue fonts represent the upregulated and downregulated genes, respectively. Ald, alanine dehydrogenase; ArgB/J, arginine biosynthesis protein B/J; BCCT, betaine/carnitine/choline transporter; GbsB, choline dehydrogenase; GcvP, glycine cleavage system protein P; McsL, large conductance mechanosensitive channel protein; MFS, major facilitator superfamily; MurC/D, mur synthetases; PGA, poly-γ-glutamate; PgsB/C, PGA synthesis protein PgsB/C; PruA, Δ^1^-pyrroline-γ-carboxylate dehydrogenase.

**Table 1 foods-11-00524-t001:** Differential gene expression related to stress response proteins and ion transporters.

Description	Locus ID	Gene	Product	Gene Expression (log_2_ FC) *		
				0% vs. 10%	0% vs. 20%	10% vs. 20%
Stress	DV527_RS05825		Universal stress protein	1.88	5.89	4.01
	DV527_RS01570		Universal stress protein	0.35	3.63	3.28
	DV527_RS05850		Universal stress protein	1.24	3.09	1.86
	DV527_RS03625		Asp23/Gls24 family envelope stress response protein	1.55	2.87	1.33
	DV527_RS10290		GlsB/YeaQ/YmgE family stress response membrane protein	2.21	2.69	0.49
	DV527_RS11920		General stress protein	0.52	1.54	1.02
	DV527_RS11570		50S ribosomal protein L25/general stress protein Ctc	0.31	−0.96	−1.27
	DV527_RS06680		Asp23/Gls24 family envelope stress response protein	−0.84	−2.23	−1.39
Osmoprotectant	DV527_RS02415		Osmoprotectant ABC transporter substrate-binding protein	−0.87	1.36	2.23
	DV527_RS05510		Osmoprotectant ABC transporter substrate-binding protein	−1.59	−0.79	0.80
Channel	DV527_RS12460		Mechanosensitive ion channel family protein	−1.75	−2.42	−0.67
	DV527_RS07575	*mscL*	Large conductance mechanosensitive channel protein MscL	−1.98	−4.73	−2.75
Symporter	DV527_RS02905	*gltS*	Sodium/glutamate symporter	0.67	1.68	1.02
	DV527_RS05020		Dicarboxylate/amino acid: cation symporter	0.47	2.04	1.57
	DV527_RS04605	*putP*	Sodium/proline symporter PutP	−3.66	−4.06	−0.40
	DV527_RS07535		Alanine:cation symporter family protein	−0.49	−1.72	−1.23
Na^+^/H^+^ antiporter	DV527_RS09405	*mnhB1*	Na^+^/H^+^ antiporter Mnh1 subunit B		−0.38	
	DV527_RS09410	*mnhC1*	Na^+^/H^+^ antiporter Mnh1 subunit C	−1.15	−1.76	−0.61
	DV527_RS09415	*mnhD1*	Na^+^/H^+^ antiporter Mnh1 subunit D	−0.25	−0.36	
	DV527_RS10700	*mnhB2*	Na^+^/H^+^ antiporter Mnh2 subunit B	−0.64	−1.28	−0.63
	DV527_RS10695	*mnhC2*	Na^+^/H^+^ antiporter Mnh2 subunit C	−1.31	−2.13	−0.82
	DV527_RS10690	*mnhD2*	Na^+^/H^+^ antiporter Mnh2 subunit D	−0.91	−1.44	−0.53
	DV527_RS10685	*mnhE2*	Na^+^/H^+^ antiporter Mnh2 subunit E	−1.86	−2.82	−0.96
	DV527_RS10680	*mnhF2*	Na^+^/H^+^ antiporter Mnh2 subunit F		−1.01	−0.51
	DV527_RS02145		Sodium:proton antiporter	−0.52	−3.42	−2.90
	DV527_RS11800		Sodium-dependent transporter	−0.66	−3.86	−3.20
Potassium	DV527_RS08810		TrkA family potassium uptake protein	−0.32		0.50
	DV527_RS09095		TrkH family potassium uptake protein	−0.27	−0.95	−0.69
Glycine betaine	DV527_RS07570		BCCT family transporter	−1.31	−2.37	−1.06
	DV527_RS07250		BCCT family transporter	−0.49	−4.89	−4.40
	DV527_RS01165		BCCT family transporter	−2.58	−5.73	−3.16
	DV527_RS01180	*betA*	Choline dehydrogenase	−1.36	−6.02	−4.66
	DV527_RS01175	*betB*	Betaine–aldehyde dehydrogenase		−5.74	−4.69
MFS transporter	DV527_RS02780		Sugar porter family MFS transporter	−0.48	2.56	3.03
	DV527_RS12535		Sugar porter family MFS transporter	−1.01	2.03	3.03
	DV527_RS02165		MFS transporter	1.45	4.96	3.51
	DV527_RS11830		MFS transporter	−0.25	0.70	0.95
	DV527_RS02855		Multidrug efflux MFS transporter	−0.64	−1.53	−0.90
	DV527_RS02475		MFS transporter	−1.61	−2.52	−0.91

* Gene expression is represented using the absolute log_2_ fold change (FC) values having adjusted *p*-values < 0.05. Red and blue colors correspond to upregulated or downregulated gene expression level, respectively.

**Table 2 foods-11-00524-t002:** Differential gene expression related to synthesis and transport for compatible solutes.

Description	Locus ID	Gene	Product	Gene Expression (log_2_ FC) *		
				0% vs. 10%	0% vs. 20%	10% vs. 20%
Alanine	DV527_RS05830	*ald*	Alanine dehydrogenase	1.01	4.05	3.04
	DV527_RS05660	*murC*	UDP-*N*-acetylmuramate-l-alanine ligase	1.04	1.74	0.69
	DV527_RS08415	*murD*	UDP-*N*-acetylmuramoyl-l-alanine-d-glutamate ligase	1.75	1.00	−0.74
	DV527_RS05740		Alanine–glyoxylate aminotransferase family protein	−0.82	0.80	1.62
Proline	DV527_RS05540		Proline dehydrogenase family protein	−0.59	4.14	4.73
	DV527_RS02405		Betaine/proline/choline family ABC transporter ATP-binding protein	−0.70	1.24	1.94
Glycine	DV527_RS06635	*gcvPB*	Aminomethyl-transferring glycine dehydrogenase subunit GcvPB	1.83	4.33	2.51
	DV527_RS06630	*gcvPA*	Aminomethyl-transferring glycine dehydrogenase subunit GcvPA	1.70	4.13	2.43
	DV527_RS06625	*gcvT*	Glycine cleavage system aminomethyltransferase GcvT	1.31	3.74	2.44
	DV527_RS11235		Glycine C-acetyltransferase	0.77	2.48	1.71
	DV527_RS00920		Choloylglycine hydrolase family protein	1.51	2.18	0.67
	DV527_RS03365		Lipid II:glycine glycyltransferase FemX	−1.04	−0.76	0.27
Glutamate	DV527_RS01735	*pruA*	l-glutamate γ-semialdehyde dehydrogenase	0.91	6.02	5.12
	DV527_RS02350		FMN-binding glutamate synthase family protein	1.55	3.60	2.04
	DV527_RS06030	*hemL*	Glutamate-1-semialdehyde 2,1-aminomutase	1.90	2.65	0.75
	DV527_RS12045		Poly-γ-glutamate hydrolase family protein	1.16	2.42	1.26
	DV527_RS07840		Poly-γ-glutamate hydrolase family protein	1.14	1.83	0.69
	DV527_RS08520	*racE*	glutamate racemase	0.27	1.52	1.25
	DV527_RS11825		Poly-γ-glutamate hydrolase family protein	2.60	0.53	−2.07
	DV527_RS11740	*gltB*	glutamate synthase large subunit	−3.21	−0.42	2.79
	DV527_RS01265	*argJ*	Bifunctional glutamate *N*-acetyltransferase acetyltransferase ArgJ	−1.38	−1.12	
	DV527_RS01910	*pgsC*	Poly-γ-glutamate biosynthesis protein PgsC		−1.75	−1.77
	DV527_RS01270	*argB*	acetylglutamate kinase	−2.14	−1.78	
	DV527_RS01905	*pgsB*	Poly-γ-glutamate synthase PgsB		−1.94	−1.52

* Gene expression is represented using the absolute log_2_ fold change (FC) values having adjusted *p*-values < 0.05. Red and blue colors correspond to upregulated or downregulated gene expression level, respectively.

## Data Availability

The data presented in this study are openly available in the Sequence Read Archive (SRA) of the National Center for Biotechnology Information Search (NCBI) database under accession code PRJNA797916.

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
