# Peer review of "Transcriptome Analysis of Halotolerant Staphylococcus saprophyticus Isolated from Korean Fermented Shrimp"

_foods, 2022, doi:10.3390/foods11040524_

Round 1

Reviewer 1 Report

In this study, the authors aim to understand the basis of halotolerance in Staphylococcus saprophyticus, an organism isolated from saeu-jeotgal, a high-salinity food. To do so, the authors chose a transcriptomic approach, by performing and comparing RNA-seq of this organism on medium with 0%, 10% and 20% NaCl. Interestingly, in addition to the expected upregulation of stress proteins in the high-salt conditions, the authors noticed upregulation of several aminoacid biosynthesis pathways, notably glutamate.

Overall, this paper is well written and presents in clear figures the results of the transcriptomic data, most notably in the Tables and in the model in Figure 5. This study will provide a useful resource for researchers interested in the field of halotolerance.

A minor potential improvement for the figures would be:

- For Table 1 and Table 2, the legend should identify the reason for excluding some values (for example mnhB1 in 0% vs 10%): is it based on effect size (such as |val|<0.3), on statistics (such as p>0.05), or on another factor (such as minimum read coverage, etc)?

- For Figure 5, a slightly larger arrowhead size would make the figure clearer. It could also help to add arrows representing import/export of solutes and ions for the symporters and antiporters where this information is known. This would allow Fig 5 to summarize in an even clearer way the general metabolic & transporter adaptation to high-salt.

Author Response

We thank the reviewers for their comments, which have improved the quality of our manuscript. Detailed responses to each comment are provided below.

A minor potential improvement for the figures would be:

- For Table 1 and Table 2, the legend should identify the reason for excluding some values (for example mnhB1 in 0% vs 10%): is it based on effect size (such as |val|<0.3), on statistics (such as p>0.05), or on another factor (such as minimum read coverage, etc)?

Author response: It is stated that if there is no log2FC value under Tables 1 and 2, it is because only values satisfying p<0.05 are displayed in the Table. FC of DV527_RS09405 (mnhB1) was -0.16 under 0% vs. 10% condition, but p value was 0.25. Similarly, DV527_RS10680 (mnhF2) also had a FC of -0.50 but a p value of 0.06. We added footnote like “Gene expression is represented using absolute log2 fold change (FC) values having adjusted p-values < 0.05.” below each Table to explain this situation.

- For Figure 5, a slightly larger arrowhead size would make the figure clearer. It could also help to add arrows representing import/export of solutes and ions for the symporters and antiporters where this information is known. This would allow Fig 5 to summarize in an even clearer way the general metabolic & transporter adaptation to high-salt.

Author response:  Figure 5 was corrected in the text as suggested. We changed the Fig. 5. 

Reviewer 2 Report

Dear Authors,

 This manuscript used transcriptomic approaches to assess a total of 831, 1314, and 1028 differentially expressed genes 17(DEGs) in the 0 vs. 10, 0 vs. 20, and 10 vs. 20% NaCl comparisons, respectively, which revealed the DEGs involved in amino acid transport and metabolism, transcription, and inorganic ion transporter and metabolism. The findings are interesting. I would recommend the publication of the manuscript. While the study presents a nice scientific experiment, logic, well designed, and the results were clearly presented, I would suggest a few minor changes:

- In the introduction, page 1, line 35-37 and 42-44: Since the authors mentioned about the unique taste and flavor of jeotgal by these microorganisms facilitating the fermentation process, producing several peptides, aromatic compounds, and functional nutrients in the preserved seafood, it would be a plus and more perfect to link those point with the key finding or result of this research. That will attract more readers in the field of basic food science.

- In the conclusion, page 11: It would be very nice if authors could give any idea of the method or this finding in food application.

Sincerely yours,

Author Response

We thank the reviewers for their comments, which have improved the quality of our manuscript. Detailed responses to each comment are provided below.

This manuscript used transcriptomic approaches to assess a total of 831, 1314, and 1028 differentially expressed genes 17(DEGs) in the 0 vs. 10, 0 vs. 20, and 10 vs. 20% NaCl comparisons, respectively, which revealed the DEGs involved in amino acid transport and metabolism, transcription, and inorganic ion transporter and metabolism. The findings are interesting. I would recommend the publication of the manuscript. While the study presents a nice scientific experiment, logic, well designed, and the results were clearly presented, I would suggest a few minor changes:

- In the introduction, page 1, line 35-37 and 42-44: Since the authors mentioned about the unique taste and flavor of jeotgal by these microorganisms facilitating the fermentation process, producing several peptides, aromatic compounds, and functional nutrients in the preserved seafood, it would be a plus and more perfect to link those point with the key finding or result of this research. That will attract more readers in the field of basic food science.

Author response:  As suggested, we linked the results and the unique taste and flavor of jeotgal. Check the Lines 312-313.

- In the conclusion, page 11: It would be very nice if authors could give any idea of the method or this finding in food application.

Author response: We added the sentence in Lines 345-347.

Reviewer 3 Report

The manuscript (foods-1587341) entitled "Transcriptome Analysis of Halotolerant Staphylococcus saprophyticus Isolated from Korean Fermented Shrimp” is a very interesting paper addressing an important, little knowledge issue to determine the mechanisms for osmoadaptation of S. saprophyticus isolated from saeu-jeotgal. Microorganisms are important factors that play dominant roles in the shrimp paste fermentation process. Jeotgals prepared by traditional methods suffer from some problems, especially long time required for completion of fermentation. Under high salinity, growth of most microorganisms is inhibited except some halophiles or halo-tolerant organism.

Another new aspect is the increased expression of genes related to glutamate, an important molecule that causes halotolerance in bacteria. This study reports on the role of glutamate as a key compatible solute and presents a potential molecular mechanism underlying the adaptive response of S. saprophyticus to salt stress. Also, other important studies Jeong et al. (2016) the analysis of the genome of S. equorum KS1039 do not record the gene code of any of the virulence factors. This trait and its high salt tolerance meet the requirements of a candidate for a high salt fermentation food starter.

These studies will provide the basis for further comparative and functional genomic analyzes and help identify which strains are suitable for use in the production of food for human consumption.

The paper is scientifically sound and worthy to be considered for publication. The experimental design is correct; the methods based on cell culture in differential salt concentration condition coupled to was investigated using RNA sequencing are correctly described are interesting tools to evaluate this condition. The statistical analysis is correct.

The manuscript is well written.

Author Response

We thank the reviewers for their comments, which have improved the quality of our manuscript.